# Paracoccidioidomycosis: What We Know and What Is New in Epidemiology, Diagnosis, and Treatment

**DOI:** 10.3390/jof8101098

**Published:** 2022-10-18

**Authors:** Paulo Mendes Peçanha, Paula Massaroni Peçanha-Pietrobom, Tânia Regina Grão-Velloso, Marcos Rosa Júnior, Aloísio Falqueto, Sarah Santos Gonçalves

**Affiliations:** 1Department of Medical Clinic, Division of Infectious Diseases, Federal University of Espírito Santo (UFES), Vitoria 29043900, Brazil; 2Department of Medicine, Division of Infectious Diseases, Federal University of São Paulo (UNIFESP), São Paulo 04039032, Brazil; 3Department of Dental Clinic, Federal University of Espírito Santo (UFES), Vitoria 29043900, Brazil; 4Department of Medical Clinic, Federal University of Espírito Santo (UFES), Vitoria 29043900, Brazil; 5Department of Pathology, Infectious Diseases Postgraduate Program, Center for Research in Medical Mycology, Federal University of Espírito Santo (UFES), Vitoria 29043900, Brazil

**Keywords:** paracoccidioidomycosis, *Paracoccidioides* spp., endemic mycosis, thermodimorphic fungi

## Abstract

Paracoccidioidomycosis (PCM) is a systemic mycosis endemic to Latin America caused by thermodimorphic fungi of the genus *Paracoccidioides.* In the last two decades, enhanced understanding of the phylogenetic species concept and molecular variations has led to changes in this genus’ taxonomic classification. Although the impact of the new species on clinical presentation and treatment remains unclear, they can influence diagnosis when serological methods are employed. Further, although the infection is usually acquired in rural areas, the symptoms may manifest years or decades later when the patient might be living in the city or even in another country outside the endemic region. Brazil accounts for 80% of PCM cases worldwide, and its incidence is rising in the northern part of the country (Amazon region), owing to new settlements and deforestation, whereas it is decreasing in the south, owing to agriculture mechanization and urbanization. Clusters of the acute/subacute form are also emerging in areas with major human intervention and climate change. Advances in diagnostic methods (molecular and immunological techniques and biomarkers) remain scarce, and even the reference center’s diagnostics are based mainly on direct microscopic examination. Classical imaging findings in the lungs include interstitial bilateral infiltrates, and eventually, enlargement or calcification of adrenals and intraparenchymal central nervous system lesions are also present. Besides itraconazole, cotrimoxazole, and amphotericin B, new azoles may be an alternative when the previous ones are not tolerated, although few studies have investigated their use in treating PCM.

## 1. Introduction

Paracoccidioidomycosis (PCM) is a systemic fungal infection caused by thermodimorphic microorganisms belonging to the genus *Paracoccidioides*. The disease is endemic to South and Central America, but imported cases have been reported in North America, Europe, Africa, and Asia. The fungus is a soil saprophyte; classically, humans get infected through agricultural activities. Thus, socioeconomic changes in Latin America in recent decades—namely, reduction in human labor in agriculture—have directly impacted the epidemiology of PCM [1,2,3,4].

Although PCM-related mortality is low, morbidity is high once the chronic form’s sequelae are present in almost 50% of patients despite treatment [5]. Often, a lack of early clinical suspicion results in delayed treatment.

A recent review of endemic mycosis in America recognized that PCM had the fewest diagnostic tools available [6]. Microbiological studies on respiratory, ganglion, or mucocutaneous samples can be easily performed with a direct microscopic examination (DME). A limitation is that expertise is required to recognize the characteristic fungal structures [6].

Commercial kits for serological diagnosis have been developed, but countries still face production, distribution, and cost problems; thus, in most centers, the antigen is produced *in-house*. Some investigations have been conducted to determine triggers that detect PCM owing to all the species of *Paracoccidioides*. Novel molecular biomarkers have been studied, but warrant further investigation to validate their use and identify the best scenarios to employ them [6,7,8,9].

Treatment-wise, *Paracoccidioides* spp. are susceptible to sulfonamides, azoles, and amphotericin B (both conventional and lipidic formulations). Although itraconazole has been proven to be more effective than cotrimoxazole to treat mild and moderate PCM, in Latin America, the last is still largely used because of its lower cost. Similarly, although the lipid formulations of amphotericin B are less toxic than the conventional one, their high price limits its use in low-income countries [10,11,12].

This paper summarizes essential knowledge and updates in the epidemiology, diagnosis, and treatment of PCM. In addition, we aimed to highlight the need for advancement in diagnostic and therapeutic strategies that will reduce the burden of PCM in Latin America.

## 2. Epidemiology of Paracoccidioidomycosis

### Species and Geographic Distribution

In recent years, the taxonomy of the genus *Paracoccidioides* has undergone notable changes. Until 2005, *Paracoccidioides brasiliensis* was considered the only species that caused PCM. In 2006, Matute et al. [13], through genotypic studies, revealed variations in *P. brasiliensis* encompassing four genetic variants (S1, PS2, PS3, and PS4) [12]. In 2009, Teixeira et al. [14] described the new species *Paracoccidioides lutzii* in the midwest region of Brazil. In 2017, Turissini et al. [15] proposed a new classification for the *P. brasiliensis* variants’ phylogenetic species (S1, PS2, PS3, and PS4), comprising four new species: *P. brasiliensis sensu stricto*, for S1, the deep split of the S1 lineage into two clades named S1a and S1b [16]; *Paracoccidioides americana* for PS2; *Paracoccidioides restrepiensis* for PS3; and *Paracoccidioides venezuelensis* for PS4. In 2020, whole genome sequencing studies confirmed *P. brasiliensis* complex reclassification into new species [17].

Regarding geographic distribution, *P. brasiliensis sensu stricto* is widespread and predominant in the southern region of South America (Brazil, Argentina, Paraguay, Uruguay, and Bolivia). *P. americana* has been identified in the same regions, but in limited cases. *P. restrepiensis* is predominant in Colombia, with cases reported in Argentina, Peru, and Uruguay. *P. venezuelensis* is dominant in Venezuela and reported in Colombia as well. *P. lutzii* is prevalent in central-west Brazil, with scattered cases outside this area [14,15,16,17,18].

Recent developments in genetic studies of *Paracoccidioides’* different species and their epidemiology show that they co-exist in several regions of Latin America, with a clear overlapping distribution. Interestingly, when the magnitude of gene flow between species was analyzed, Mavengere et al. [19] confirmed that *Paracoccidioides* species rarely exchange genes despite extensive geographic overlap [16,17,18,19].

Although the different species have shown differences in conidial morphology and antigenic display, no recent studies have shown clear implications of species diversity on disease clinical manifestation and treatment [20,21,22]. Finally, recent genetic analyses of specimens from dolphins with lobomycosis placed the DNA sequences of *Lacazia loboi* within *Paracoccidioides* species, proposing the taxonomy of the dolphin pathogen as *Paracoccidioides cetii*, and the human pathogen as *Paracoccidioides loboi* [23].

In Latin America, Brazil has the highest number of PCM cases (80%), followed by Colombia, Venezuela, Ecuador, and Argentina [2]. The disease has been reported as far north as Mexico and south as Argentina [24,25]. No cases have been reported in Nicaragua, Belize, most Caribbean islands, Guyana, Surinam, or Chile [2,26].

Over 100 PCM cases have been reported in Europe, the United States of America, Canada, Africa, and Asia. The patients are usually non-autochthonous, mainly immigrants or travelers from endemic countries [27,28,29]. In a recent systematic review, Wagner et al. [4] identified 83 patients with PCM in 11 European countries, most of whom were from Spain, Italy, and Germany. The patients were mostly men aged 23–83 years, with a latency period of 6 days to 50 years [4]. In Asia, specifically in Japan, 16 imported cases of PCM—mainly from Brazil—have been reported [30,31,32,33,34,35,36,37,38]. Figure 1 shows all countries with endemic PCM and imported cases.

A major risk factor for PCM is soil exposure in rural areas, being an occupational disease for farmers in endemic regions [2,3]. Person-to-person transmission has not yet been documented [1]. In terms of personal and social history, smoking has been elicited in the histories of 90% of patients with chronic PCM; smokers have a 14-fold higher risk of developing the disease than nonsmokers [39,40]. Alcohol intake also increased the risk of chronic PCM by 3.5 times [2,39].

PCM’s chronic form presents in adults from 30 to 60 years, with a male predominance (ratios of 15:1 to 22:1 in Brazil) [41,42]. The acute form of the disease shows an equal distribution between sexes, especially in childhood [41]. The predominance of chronic PCM in men can be explained by the protective effect of estrogen in women, which inhibits the transformation of conidia in yeast cells after menarche [43].

The incidence of PCM in Brazil ranges from 1 to 4 per 100,000 inhabitants per year in established endemic areas in the southeastern and southern regions. In areas of recent colonization in the northern region (Amazon), the incidence rises to 9–40 cases per 100,000 inhabitants, illustrating a hyperendemic pattern in the last few decades [2,44,45]. The prevalence rate of PCM infection was 45.8% in endemic rural areas in southeast Brazil according to surveys based on skin tests, with Colombia and Ecuador having rates of 45% and 41%, respectively. Conversely, in Argentina and Venezuela, the rates were as low as 7.8% and 10.2%, respectively [39,46,47,48,49]. Notably, since PCM does not require compulsory notification in most Latin American countries, its incidence and prevalence may be underreported [6].

The most extensive study on PCM mortality performed in Brazil by Coutinho et al. reported a mortality rate of 1.45 per million inhabitants, with a total of 3.181 attributed deaths per million inhabitants between 1980 and 1995 [50]. Currently, the mortality rate is decreasing in Brazil’s southeastern and southern regions, and increasing in the north. The state of Paraná, which had the highest mortality rate for PCM in south Brazil (3–4.29 deaths per million inhabitants during 1980–1995), showed a reduction in the average annual mortality rate to 1.17 deaths per million inhabitants during 2007–2020 [51]. Contrastingly, the state of Rondônia in the northern region drastically progressed from a mortality rate of 3.65 per million inhabitants in 1980–1995 to 8.2 in 2002–2004 [44,52].

Death from PCM is due to disseminated disease, respiratory insufficiency, and adrenal insufficiency. These three are frequently associated with the chronic form of the disease and may develop long after antifungal treatment has been completed [41,53,54]. Despite the indolent course of the disease and the availability of curative medications, difficulty accessing diagnostics and treatment contributes to an unfavorable prognosis [3,50].

The epidemiology of PCM has significantly changed in terms of frequency, demographic characteristics, and geographic distribution, owing to human migration, environmental interventions, and climate change. The opening of new agricultural frontiers and deforestation have contributed to the rise in the incidence of PCM in northern Brazilian states. In contrast, the reduction in child labor in rural areas and mechanization of agriculture have decreased the number of new cases in the southern region [3,44,55,56]. An increase in *Paracoccidioides* spp. infections has been associated with the construction of the Yacyreta hydroelectric plant in northeastern Argentina [57]. Clusters of acute/subacute cases of PCM were reported after the El Niño phenomena in 1982/83 in southeast Brazil and 2009 in northeast Argentina [55,58], as well as after the massive land removal during the construction of the ring road in the Rio de Janeiro metropolitan area in 2016 [59]. These clusters underscore the urgent need for the surveillance of new cases in endemic regions undergoing climate change and human interventions, such as deforestation and massive constructions, to ensure early diagnosis and treatment [59].

## 3. Diagnosis of Paracoccidioidomycosis

### 3.1. Clinical Diagnosis

The first challenge in diagnosing PCM is to think about the disease when, even in endemic countries and more so in countries outside Latin America, physicians are unfamiliar with this systemic mycosis. Consequently, late diagnosis increases morbidity and mortality rates [4,40].

PCM infection is acquired by inhaling fungal propagules found in the environment, but only 1% to 2% of infected individuals will develop clinical manifestations during their lives [3]. Those who remain asymptomatic can control fungal replication through a robust Th-1 immune response pattern, characterized by cytokine release that activates macrophages and TCD4+ and TCD8+ cells, forming compact granulomas. This stage is named PCM infection [60]. Among those individuals who progress from infection to illness, 5–25% present with the acute/subacute clinical form of PCM, in which Th-2 and Th-9 immune response patterns activate B-lymphocytes that produce high levels of antigen-specific IgA, IgE, and IgG4 [61]. The other 75–95% of cases will evolve from the latent stage to the chronic form of the disease many years later, usually after the fourth decade of life [3].

Chronic PCM manifests gradually and may occur years after exposure to *Paracoccidioides* when the patient is already living in urban areas or outside endemic regions [3]. This form mainly affects the lungs; mucous membranes; skin; and, eventually, the adrenal and central nervous systems [2,41,53]. The main manifestation of chronic PCM in approximately 90% of patients is pulmonary, with symptoms of cough, dyspnea, and sputum expectoration [3,41]. A recent study by Dutra et al. [62] in a southeastern Brazilian hospital found that 59.6% of the patients with PCM had granulomatous ulcerated oral lesions. The oral lesions (Figure 2A,B) may be the first visible physical manifestation of the disease noticed by the patient, and may lead to a prompt diagnosis. However, even with adequate treatment, patients may develop the residual form of PCM, owing to fibrosis of the affected organs [3,5].

The acute/subacute form of the disease shows rapid and disseminated progression in the form of skin lesions; lymphadenopathy; and eventual suppuration, fever, and anorexia (Figure 2C) [3,41,63]. This form characteristically develops a few weeks or months after fungal exposure [2].

In immunocompromised patients, a mixed clinical form of PCM, with characteristics of both chronic and acute forms of the disease, has been observed. Pulmonary involvement can coincide with generalized adeno- and hepatosplenomegaly. In patients with the mixed form, multiple, exuberant skin involvement; lytic bone lesions; and central nervous system involvement can be present, indicative of severe disease (Figure 2D). Patients with HIV co-infected with PCM make up the bulk of immunosuppressed patients with PCM; however, cases of PCM in transplant patients and those receiving immunobiological therapy have also been reported [63,64,65,66].

PCM is often confused with tuberculosis in Latin America, owing to its high prevalence and the similar clinical presentations of both diseases [34]. Besides tuberculosis, the most relevant differential diagnoses of chronic pulmonary PCM are other fungal infections, such as coccidioidomycosis and histoplasmosis. Sarcoidosis, pneumoconiosis, and interstitial pneumonitis should also be considered. Moreover, it is essential to rule out concomitant diseases. Tuberculosis and PCM can occur simultaneously or sequentially in 5.5–19% of cases [41,53,67,68]. Additionally, PCM and solid cancers of the respiratory and gastrointestinal tracts share similar risk factors (male sex, smoking, and alcohol intake). Solid neoplasias and *Paracoccidioides* infection have been shown to co-exist in 0.16–11% of patients [69].

In patients with mucocutaneous PCM, the differential diagnosis should include leishmaniasis, tuberculosis, chromoblastomycosis, leprosy, syphilis, and neoplasia. In individuals with acute PCM, clinicians should be concerned about hematologic neoplasms, histoplasmosis, tuberculosis, and visceral leishmaniasis [3]. It is important to remember that many infectious diseases share the same endemic areas as PCM.

Finally, patients with chronic pulmonary PCM are at higher risk of more severe illness with COVID-19 coinfection. Despite that, during the pandemic, only one case of SARS-CoV-2 and *Paracoccidioides* spp. co-infection was described in an individual with the acute form of PCM. The scarcity of documented cases probably reflects underdiagnosis or underreporting in Latin American countries that had their health systems overwhelmed by COVID-19 [70,71,72].

### 3.2. Laboratory Diagnosis

Laboratory diagnosis via microscopy remains the gold standard method for diagnosing PCM. This may show the presence of the etiologic agent in biological fluids and tissue sections or the isolation of the fungus from clinical specimens, owing to the characteristic appearance of typical *Paracoccidioides* spp. yeast forms [73]. Other tools, such as cultures, immunodiffusion assays, and polymerase chain reaction (PCR) tests [7,8,9,74,75], are also used. Different types of clinical samples may be collected for testing. Mucocutaneous scrapings, sputum, bronchoalveolar lavage (BAL), cerebrospinal fluid (CSF), lymph node aspirate, biopsy, and tissue samples are those most frequently collected [76,77]. Prior processing of some samples is needed to increase their sensitivity to detection methods, including centrifugation (for sputum, BAL, CSF, and lymph node aspirate) and maceration of fragmented, biopsied tissues [7]. The sputum sample should be prepared with potassium hydroxide, sodium hydroxide, and N-acetyl-L-cysteine before being added to a suitable culture medium at 25 °C [73]. Figure 3 shows the main diagnostic tools employed for laboratory diagnosis of PCM. 

#### 3.2.1. Mycological Diagnosis

This method includes visualizing fungal elements through DME, followed by isolation of the agent in culture media [25]. DME of the sputum, BAL, CSF, lymph node aspirate, and mucocutaneous scraping are prepared with the addition of 10–20% KOH or calcofluor, making it possible to visualize *Paracoccidioides* spp. in their parasitic form with multiple budding cells (blastoconidia) surrounding it, connected by short cellular bridges (Figure 3B,D) [78,79,80,81,82]. However, it is important to mention that for biopsy and tissue samples, the slides are mounted using 40% KOH [78]. *Paracoccidioides* cells have a thick mucopolysaccharide wall with a double-contour appearance that is birefringent under light microscopy [83,84]. *Paracoccidioides* structures resembling a “ship’s wheel” or “Mickey Mouse” are deemed pathognomonic findings [9,79,82].

The size and multiple budding distinguish *Paracoccidioides* spp. from other fungi. Nevertheless, *Paracoccidioides* isolates can be mistaken for *Histoplasma capsulatum* and *Cryptococcus* spp. when it produces small, non-budding cells, and when *Cryptococcus* does not produce its capsule efficiently [84,85].

Moreto et al. [7] evaluated the diagnostic methods for PCM at a university hospital between 1976 and 2004. They observed that in the DME of 51 different tissue specimens and 112 sputum samples, the sensitivity was 75% and 63%, respectively. For 483 sputum cell blocks, the values found were 55%. Since the PCM chronic form is the most frequent form encountered, sputum is the biological material most commonly evaluated under DME, and the sensitivity will depend on the processing method of that material [40,53,86]. Although DME is a simple, fast, and low-cost technique implemented in small laboratories, its sensitivity is low [40,45,53,86,87]. Thus, DME cannot provide a conclusive diagnosis in cases of negative results. Owing to the heterogeneity of the sample fractions, DME can mistakenly lead to the assumption that the fungus does not exist in the entire sample [75].

#### 3.2.2. Cultures

*Paracoccidioides* take an average of 3–6 weeks to grow on fungal culture media [88,89,90]. Nonetheless, the growth of this fungus varies in the different studies, with a sensitivity of 80 to 97% of cases [21,90,91,92]. The results should be evaluated about 4 weeks after the cultivation of the sample at 25 °C. To reduce the running cultivation time, samples must be cultured simultaneously at 25 °C and 37 °C. The culture media most frequently used are Sabouraud, Mycosel, and Fava-Netto agar. Other media, such as mycobiotic agar, brain heart infusion agar (BHI), Sabouraud dextrose plus BHI broth (SABHI), agar-yeast extract-phosphate, agar-yeast extract, agar-yeast extract-penicillin plus streptomycin and cycloheximide, and Kelley medium with hemoglobin, are also employed [75,78,79,84,93]. According to Hahn et al. [78], the successful recovery of *Paracoccidioides* from clinical specimens depends on several factors, including the culture media and the number of tubes or plates seeded, besides the decontamination with antibiotics of sputum and bronchial lavage fluid. Despite its slowness, culture is always recommended because it allows species identification using molecular biology [9,94,95,96].

#### 3.2.3. Histopathological Diagnosis

Histopathological examination is a valuable tool (≥95% sensitivity) for PCM diagnosis. It can also determine disease severity [7,53,97]. The sections can be stained with hematoxylin/eosin (H & E), Grocott’s methenamine silver, and periodic acid–Schiff (PAS). These last two are specific stains that increase sensitivity [62]. When stained with H&E, an inflammatory response can be observed in the parasite–host interaction. Organizing granulomas or a combination of suppurative and granulomatous infiltrates can also be seen [7,75,98]. If the samples are sufficient, DME and tissue culture can be performed [86]. It is worth mentioning that biological samples for histological examination are fixed in formaldehyde solution. For culture, the biopsy and tissue samples must be placed in a sterile container under a sterile physiological saline solution.

#### 3.2.4. Immunological Diagnosis: Antibody Detection

In 1916, Arthur Moses, an assistant physician at the Oswaldo Cruz Foundation, isolated an antigen from *P. brasiliensis*-infected cells used in a complement fixation (CF) test to diagnose PCM [20]. Since then, numerous antigens and antibody-based serological assays have been developed as alternative methods for detecting fungal structures in biological fluids/tissues and disease monitoring [99,100]. Double immunodiffusion (DID), counterimmunoelectrophoresis (CIE), immunofluorescence, radioimmunoassay, enzyme-based immunoassays (ELISA), immunoblotting, dot immunoassay, western blotting, and latex particle agglutination (LA) are some important techniques in use [9,99,100,101,102,103,104,105,106,107,108]. Though many validated methods for detecting anti-*Paracoccidioides* serum antibodies exist, most are conducted only in research centers. There also remains significant limitations, owing to cross-reactivity with other infectious fungi, such as *Histoplasma* and *Aspergillus* [25,99,109].

DID is the most widely used method for detecting anti-*Paracoccidioides* serum antibodies in endemic countries [99,110]. Its advantages include the ease of performing the quantitative techniques, low cost, and high specificity (85–100%) [34,53,89,90]. However, its sensitivity can range from 80 to 95% depending on the antigen applied [111,112]. CIE has similar specificity to DID (>95%) with slightly higher sensitivity (77–100%). Both techniques are recommended as serological screening tests for patients with suspected PCM because of their faster turnaround time than microbiological methods [25,99,111,113,114].

By applying the immunoblotting technique, de Camargo et al. [115] assessed several exoantigens produced by *P. brasiliensis* isolates against serum from PCM-positive patients. IgG anti-*P. brasiliensis* was discovered in some cell surface components, but the most promising were glycoproteins gp70 (70 KDa) and gp43 (43 KDa). The latter is commonly recognized by IgG antibodies, and is reactive in 100% of patients with PCM caused by *P. brasiliensis sensu stricto*.

Several components with antigenic ability to distinguish circulating antibodies in patients with suspected PCM have been tested. Glycoprotein gp43 is the most commonly used component that can be presented as a cell-free antigen (CFA), exo-antigen (ExoAg), and recombinant or purified antigen [9,110,115,116]. Throughout the exponential growth phase of *P. brasiliensis*, its cells secrete this antigen, found in almost all isolates [75]. However, there is decreased expression of this antigen in patients infected with *P. lutzii*, and false-negative results may result [9,110,115,116]. These differences in antigenic composition are probably related to phylogenetic peculiarities [117]. In addition, gp43 may trigger cross-reactivity in patients with histoplasmosis or lobomycosis because its epitope is a galactose-containing carbohydrate, common among pathogenic fungi [75,118].

By evaluating different antigenic preparations from *P. lutzii* using the immunodiffusion technique, Gegembauer et al. [112] demonstrated that tests employing *P. brasiliensis* antigens might yield false-negative results when *P. lutzii* is the causative agent [109,119]. Maifrede et al. [109] showed that 7 of 21 sera samples negative for *P. brasiliensis* antigen were positive for *P. lutzii* when the Pb339 exoantigen and PIEPM208 CFA were applied. We can infer that the frequency of *P. lutzii* may be higher than reported in endemic areas because gp43 is the most commonly used antigen in routine laboratory examinations [9,77,109].

In 2021, an American company began commercializing a DID-based test to detect *Paracoccidioides* serum antibodies (ID Antigen^®^). In the same year, Cocio and Martinez [110] used CIE and DID to evaluate the sensitivity and specificity of the antigen in ID Antigen^®^. They found that of the 24 PCM-positive serum samples of patients with active PCM, 100% were reactive in CIE methodology using ID Antigen^®^, including 11 cases of infection by *P. brasiliensis sensu stricto*, one by *P. americana* and one by *P. lutzii*. The test’s specificity was 100%, with negative results for histoplasmosis, aspergillosis, and other diseases, and an overall 75% sensitivity with PCM sera. Therefore, the antigen available in the commercial test could diagnose PCM caused by three different species.

Considering that five *Paracoccidioides* species have been recognized as PCM agents in endemic areas, new antigen preparations must be, and are, being investigated to expand the use of PCM serology with increased sensitivity and specificity.

#### 3.2.5. Antigen Detection

Antibody detection is not the best choice for all patients with PCM. To illustrate, in many studies, immunocompromised patients and those with severe forms of acute/subacute disease showed decreased antibodies, with half showing none [84,120,121]. Therefore, identifying antigens instead in these cases would be more suitable.

In 1997, using the inhibition ELISA technique (inh-ELISA), Colombian researchers discovered a monoclonal antibody to detect the 87 kDa antigen in patients with PCM. In patients with the disease, acute, multifocal, and unifocal forms were detected in 100%, 83.3%, and 60% of patients, respectively [122]. Since then, several antigenic molecules and tools have been characterized and evaluated [74,78,105,106,107,108]. Notably, anti-gp43 and anti-gp70 monoclonal antibodies are still the most frequently investigated glycoproteins [86]. However, cross-reactions have been obtained with heterologous sera, such as sera from patients with aspergillosis, cryptococcosis, and histoplasmosis [96].

Xavier et al. [106] analyzed the Platelia™ *Aspergillus* enzyme immunoassay (EIA) (Bio-Rad, Marnes-la-Coquette, France) as a diagnostic tool for 30 PCM patients and found a positivity rate of 50%. This method is widely used to detect galactomannan in patients suspected of having invasive aspergillosis [123].

Recently, Melo et al. [108] investigated the performance of (1,3)-β-D-glucan assays (BDG), a test used to diagnose invasive fungal infections in patients with PCM. Fifty-two serum samples from 29 patients with acute and chronic PCM were evaluated. Despite its excellent diagnostic sensitivity (96.5%), it did not contribute to disease monitoring.

Some commercial methods have been validated and are currently available for detecting specific fungal antigens in patients with cryptococcosis, histoplasmosis, coccidioidomycosis, blastomycosis, aspergillosis, and candidiasis. However, progress has not been made on making commercial tests for detecting *Paracoccidioides* antigens available. It is essential to highlight that if the antigen tests were commercially available, they could make the serological diagnosis of PCM more accessible to patients who live far from referral centers.

#### 3.2.6. Molecular Detection

In the last century, molecular tools have provided crucial information for the taxonomic classification and epidemiological, diagnostic, and therapeutic management of pathogenic fungi. Several molecular methods, including PCR-derived techniques, have opened doors for the early diagnosis of fungal diseases and the identification of etiologic agents [9]. PCR, loop-mediated isothermal amplification (LAMP), quantitative real-time PCR (qPCR), nested and semi-nested PCR, and duplex PCR-assay have been found to detect *Paracoccidioides* genetic material directly from clinical samples [7,9,77,124,125,126,127,128]. Most assays are based on primary markers, such as the *GP43* gene and the internal transcribed spacer (ITS) region of ribosomal DNA [77].

In 2021, Pinheiro et al. [77] developed a duplex PCR single-assay capable of detecting and differentiating members of the *P. brasiliensis* complex and *P. lutzii* from paraffin-embedded tissue blocks [62]. This methodology became vital in clinical laboratory practice, particularly in diagnosing atypical cases, such as those with seronegative yet positive DME results, and in examining patients with co-infections.

Despite having similar sensitivity and specificity as DME and histopathological techniques, molecular methods might have a better yield with materials with a low burden of infection (serum, BAL, CSF), and may be more sensitive than DID. However, the molecular approach is performed based on *in-house* tests, for which currently external quality assessments are lacking [7,77,128]. Moreover, using molecular techniques for diagnosing disease-causing fungi directly from the clinical sample is challenging because of the complexity of DNA extraction. In addition, databases with genome sequences for these microorganisms are under construction. The very nomenclature of these agents requires continuous updating in the laboratory. In other words, the molecular tools, to be successfully used, have to adapt to the objectives of the study. Knowledge is constantly evolving in the study of fungi, and these techniques are not yet validated for use in the routine diagnosis of PCM. Furthermore, it is worth emphasizing that molecular tests are expensive, and most of the population affected by PCM belongs to developing countries.

## 4. Diagnostic Imaging

The conventional imaging of different systems affected by the fungus can contribute to PCM diagnosis and help identify the disease’s acute, chronic, or sequel forms. The most common findings in PCM are pulmonary lesions, for which chest radiography or computed tomography (CT) can be used, with the latter being more sensitive to abnormalities. Ultrasonography, CT, or magnetic resonance imaging (MRI) can be used to examine the abdominal region. For the head and neck, CT or MRI are used; and for examining the osteoarticular system, radiography, CT, or MRI are acceptable options [115]. Recently, Cunha et al. [129] studied the effectiveness of F-fluorodeoxyglucose-positron emission tomography (FDG-PET/CT)/CT) in evaluating the extent of active disease in patients with PCM under antifungal treatment, and demonstrated that FDG-PET/CT could help detect active lesions and is more sensitive than conventional imaging methods.

In the lungs, which are more affected in the chronic form of the disease, the involvement is usually bilateral and symmetric, with lesions occurring mainly in the periphery and middle-third regions (Figure 4). There are multiple CT presentations, including consolidations, ground-glass opacities, nodules, masses, and cavitations (Figure 5); interlobular septal thickening is the most common finding. In patients from endemic areas, the so-called “butterfly wing” pattern and bilateral symmetric opacities in the middle region of the lungs are suggestive of PCM [130]. The “reversed halo” sign on CT, defined as a focal and round area of ground-glass opacity surrounded by a complete or nearly complete consolidation ring, can be observed in up to 10% of cases [34]. However, it is not specific to this disease. It also occurs in patients with tuberculosis, mucormycosis, invasive pulmonary aspergillosis, *Pneumocystis carinii* pneumonia, organizing pneumonia, granulomatous polyangiitis (Wegner), lymphomatoid granulomatosis, sarcoidosis, and lepidic predominant adenocarcinoma, among others [130]. Pleural effusion and pneumothorax, also manifestations of PCM, are rare findings in patients with the chronic form. There is significant heterogeneity in pulmonary presentations. Fibrotic lesions can develop in patients and be visualized on CT as architectural distortions, traction bronchiectasis, honeycomb lesions, thickening of the alveolar and interlobular septum, paracicatricial emphysema, and parenchymal bands [130]. There are usually no pulmonary parenchymal lesions in the acute form of the disease, frequently seen with pleural effusion or lymph node enlargement (37%) [130].

The abdomen is less frequently affected than the chest. Abdominal involvement is mainly observed in patients with the acute form of PCM, which may primarily involve the adrenal glands, liver, spleen, lymph nodes, and intestinal loops. The adrenal gland is the most commonly affected abdominal organ. There may be diffuse enlargements of the gland with heterogeneous attenuation on CT and MRI, with peripheral contrast enhancement in the acute form of the disease. Adrenal atrophy and calcification are usually observed during the chronic phase (Figure 6). Differential diagnoses should include other granulomatous infections, such as histoplasmosis and tuberculosis; old hemorrhage; and neoplasms, such as lymphoma, primary tumors, and metastases [131].

The literature demonstrates a wide range of involvement of the central nervous system, ranging from 1 to 27% in different case series. The involvement can be intraparenchymal, (rarely) meningeal, or a mix of both. Parenchymal involvement usually presents as hypointense lesions on T2WI (T2-weighted imaging) MRI sequences, with peripheral enhancement after contrast (Figure 7). Occasionally, the lesions may emit iso- or hyperintense signals on T2WI, and variable, but predominantly hypointense, signals on T1. Diffusion restriction mimicking a brain abscess may also be observed.

Contrast enhancement can be ring-shaped, nodular, or heterogeneous [132]. Rosa et al. [133] demonstrated the presence of a “double halo” sign on a susceptibility-weighted imaging (SWI) MRI sequence. This finding can contribute to imaging-based diagnosis for patients with suspected PCM. These investigators also demonstrated the association between PCM and mesial temporal sclerosis, known as dual pathology, in patients with epilepsy. The meningeal form commonly presents as leptomeningeal contrast enhancement. Spinal cord involvement is rare and difficult to differentiate from neoplastic lesions [134]. Isolated involvement of the osteoarticular system is rare and, when present, is part of a multisystem process. Osteolytic lesions with cortical destruction are generally observed, especially in the metaphyses and epiphyses of long bones, without periosteal reactions [135].

In summary, the most recent advances in diagnostic imaging were FDG-PET/CT for evaluating the extent of active disease in patients with PCM undergoing antifungal treatment, demonstrating that it is more sensitive than conventional imaging. Additionally, the description of the presence of the “double halo” sign on SWI MRI sequences can help diagnose neuroparacoccidioidomycosis.

## 5. Treatment

The treatment of PCM has evolved in the last decades, and the current antifungal options are cotrimoxazole, triazoles, and amphotericin B. However, there is a lack of robust trials, and current guidelines are primarily based on non-comparative studies, expert opinions, and a couple of studies that compared itraconazole and cotrimoxazole [3,10,136].

Itraconazole (ITZ) is the first therapeutic choice for mild-to-moderate presentations of PCM, with response rates of 85–90% and an adequate tolerance profile [3,10,11,36]. Currently, the drug is available in almost all Latin American countries, but with a limited presence in non-referral hospitals and at a high cost [6]. Moreover, intravenous formulations of itraconazole are not easily available, and the oral absorption of capsules is erratic, leading to unpredictable supratherapeutic or subtherapeutic plasma levels [12]. To overcome this limitation, a novel formulation of ITZ, labeled SUper BioAvailable (SUBA)-itraconazole, was developed, with a relative bioavailability of 180% when compared with conventional itraconazole (C-ITZ), and an absolute bioavailability up to 90%. Indeed, some studies suggested less variability of SUBA compared with C-ITZ capsules, especially under fasted conditions, but a recent open-label comparative trial of 160 mg SUBA bid versus 200 mg C-ITZ bid for the treatment of endemic mycosis showed almost identical serum levels with similar specific adverse events. Of note, PCM cases were not included in the trial [137,138].

Sulfanilamides were the first agents used to treat PCM in the 1940s, and, until now, cotrimoxazole has been widely used in South America to treat mild and moderate forms of PCM, owing to its lower cost than that of itraconazole [11]. It also has the advantages of oral and venous formulations, good absorption with predictable serum levels, and fewer drug–drug interactions than azoles [10].

The duration of treatment may vary from 9 to 18 months with itraconazole, and 18 to 24 months with cotrimoxazole [3]. No in vitro or in vivo evidence suggests that the different species of *Paracoccidioides* require modifications in doses of antifungal agents or the duration of treatment [21].

Severe and disseminated forms should be treated with amphotericin B for 2–4 weeks until clinical stabilization and then transitioned to the maintenance of oral treatment. Lipid formulations are preferred (3–5 mg/kg/day) because they are less toxic than deoxycholate. Unfortunately, its availability is limited in Latin America [6,73,139]. If amphotericin B is contraindicated, high-dose intravenous cotrimoxazole (800 mg/160 mg every eight hours) may be an alternative [11].

Second-generation triazoles (voriconazole, posaconazole, and isavuconazole) have not been used extensively to treat PCM. Their high cost still prevents their large-scale use, but they are expected to be potential substitutes to itraconazole when it is not tolerated [140,141]. Notably, most endemic countries from Central and South America reported limited access to posaconazole and isavuconazole [6].

A significant challenge in effectively treating PCM is the need for prolonged antifungal use, leading to poor compliance [12]. Additionally, despite long-term treatment, the sequelae, due to chronic inflammatory processes and fibrosis, may profoundly impact organ function, and are not entirely resolved after antifungal medication [136,142]. Persistent fungal antigen stimulation and immune system activation can lead to fibrosis through excessive extracellular matrix component deposition and alterations in the tissue-scarring process, despite appropriate treatment [67,143].

Strategies have been developed to reduce lung damage. Vaccination with an immunogenic recombinant antigen of *P. brasiliensis* has shown the potential to attenuate fibrosis [144,145]. Experimental animal models treated with antifungal drugs and other antibiotics or immunomodulatory compounds (itraconazole + pentoxifylline and cotrimoxazole + azithromycin) have also achieved good results [146,147]. A monoclonal antibody specific to neutrophils (mAb-anti-Ly6G), associated with itraconazole, reduced pro-inflammatory cells, fungal load, and pro-inflammatory cytokines in mice [143,148]. In addition, a recent biological therapy comprising a *P. brasiliensis* cell wall glycoconjugate monoclonal antibody (mAbF1.4) combined with cotrimoxazole showed promising results in reducing the pulmonary fungal burden in mice [149]. Additionally, mesenchymal stem cell transplantation in combination with antifungals has been found to attenuate the inflammatory response and fibrosis induced by *P. brasiliensis* [150]. Further trials in human subjects are needed to determine the effectiveness of the new therapies for PCM.

## 6. Conclusions

PCM is undoubtedly a public health problem in Latin America, where it is an occupational disease in rural populations. Indeed, despite the mechanization of agriculture reducing human exposure to the fungus in classically endemic regions, the expansion of agriculture frontiers has led to the rise of new cases of PCM among rural workers, especially in the Amazon region. Simultaneously, reports of clusters of the acute form of the disease suggest that climate changes and anthropic environmental interventions may be risk factors for PCM development once they expose individuals to high fungal burdens, even away from rural areas. Given the increase in the number of transplants performed in endemic areas, and in the indications for the use of immunobiological therapy, a growing number of immunosuppressed people are at risk of developing PCM. In non-endemic countries, when there is a clinical suspicion of infection with *Paracoccidioides* spp., it is crucial to consider individual travel and migration history up to decades ago if the chronic form of the disease is to be considered.

Improving patients’ quality of life after PCM treatment depends on early diagnosis and new therapeutic strategies to reduce pulmonary sequelae. Public health policies aiming to spread knowledge of the disease among general practitioners in rural areas of Latin America are crucial. 

In summary, efforts must be combined to achieve advances in new biomarkers for diagnosing PCM and determining the causative species. This will allow us to know more about these microorganisms’ molecular epidemiology, since culture media recovery is slow and not always possible. Moreover, this process reinforces the necessity to develop and standardize new antigens for serology diagnosis when differences in antigenic composition are probably related to phylogenetic peculiarities.

In addition, efforts must be made to ensure access to itraconazole throughout Latin America, besides optimizing antifibrotic treatments and pulmonary rehabilitation.

## Figures and Tables

**Figure 1 jof-08-01098-f001:**
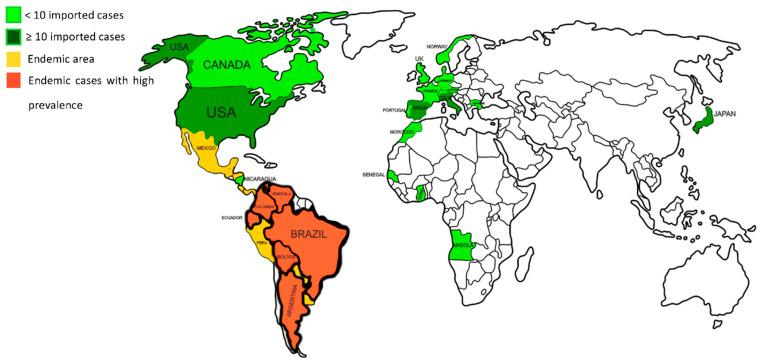
Geographic distribution of Paracoccidioidomycosis in relation to endemic areas and imported cases.

**Figure 2 jof-08-01098-f002:**
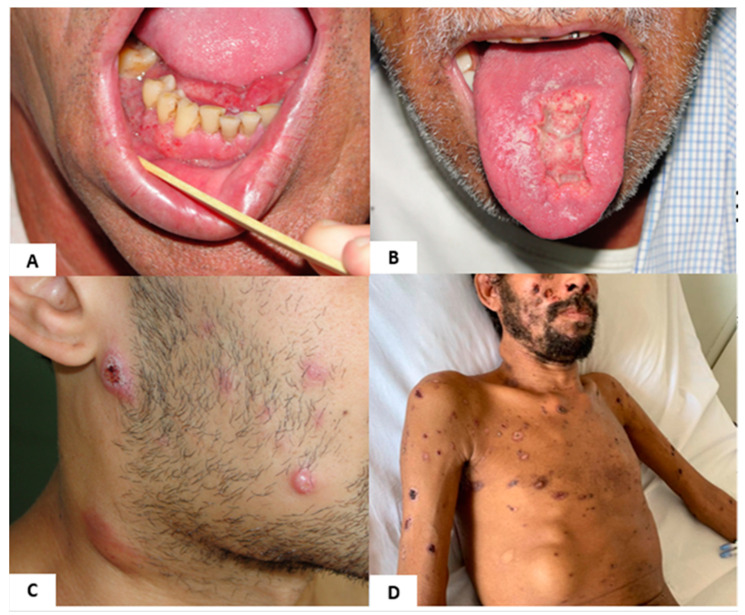
(**A**) Chronic form: gingival and lingual frenulus “mulberry-like” ulcers with hemorrhagic dots; (**B**) chronic form: deep ulcerative lesion on the tongue with infiltrative borders, hemorrhagic dots, covered with fibrin; (**C**) acute/subacute form: multiple polymorphic (nodular, papular, and ulcerated) skin lesions and cervical inflammatory lymphadenopathy; (**D**) disseminated form in a patient with PCM and HIV co-infection; the following may be seen: cervical lymphadenopathy, exuberant ulcerated, crusted skin lesions, and larges subcutaneous abscesses in the thorax and abdomen.

**Figure 3 jof-08-01098-f003:**
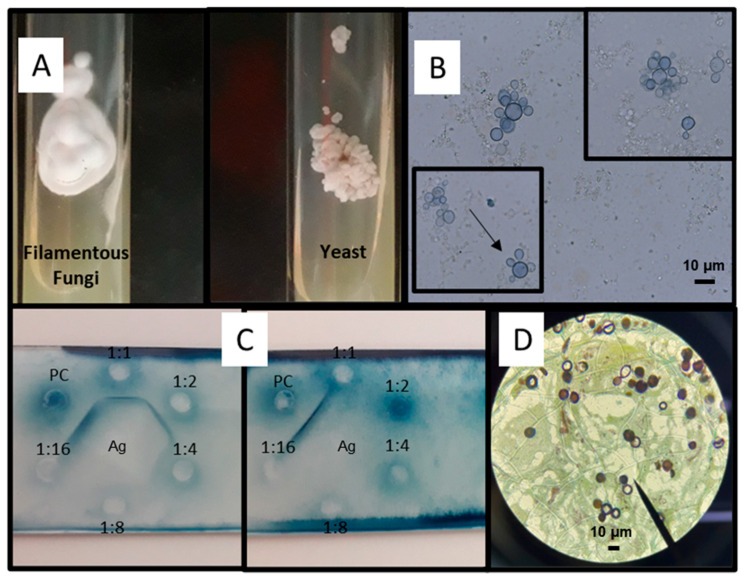
Laboratory diagnosis of Paracoccidioidomycosis. (**A**) *Paracoccidioides* spp. on Sabouraud agar slants for 14 days at 25 °C (left, filamentous phase) and 37 °C (right, yeast phase); (**B**) yeast form showing multiple budding (5 to 15 µm) seen upon direct examination of a lymph node aspirate, stained with KOH and Parker ink; Bars 10 µm (**C**) double immunodiffusion assay: *Paracoccidioides brasiliensis* exoantigen (Ag) is in the center well; sera sample from a patient with paracoccidioidomycosis is used in different titrations (wells 1:1 to 1:16) and positive control (PC); positive 1:2 (**left**) and negative (**right**); (**D**) Grocott’s methenamine silver stain showing *Paracoccidioides* yeast cells. Bars 10 µm.

**Figure 4 jof-08-01098-f004:**
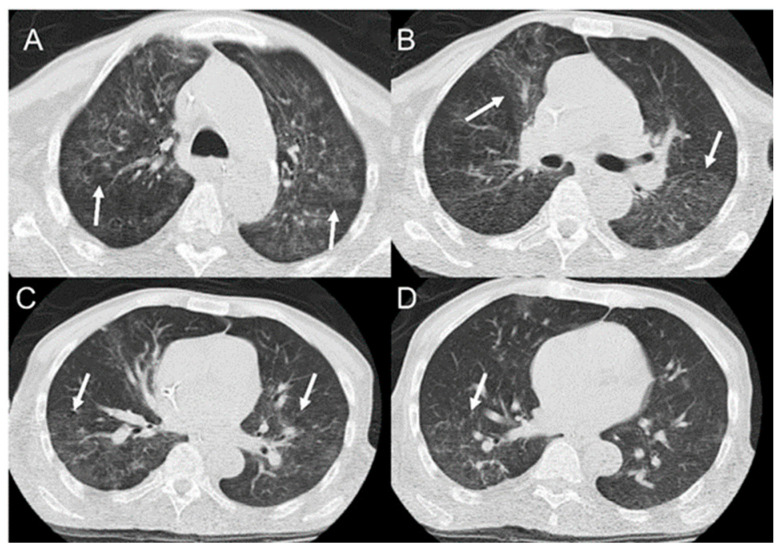
Computed tomography showing ground-glass opacities (arrows) with bilateral and symmetric distribution, occurring mainly in the periphery and the middle-third of the lungs (“butterfly wing” pattern), as shown from top to bottom in (**A**–**D**).

**Figure 5 jof-08-01098-f005:**
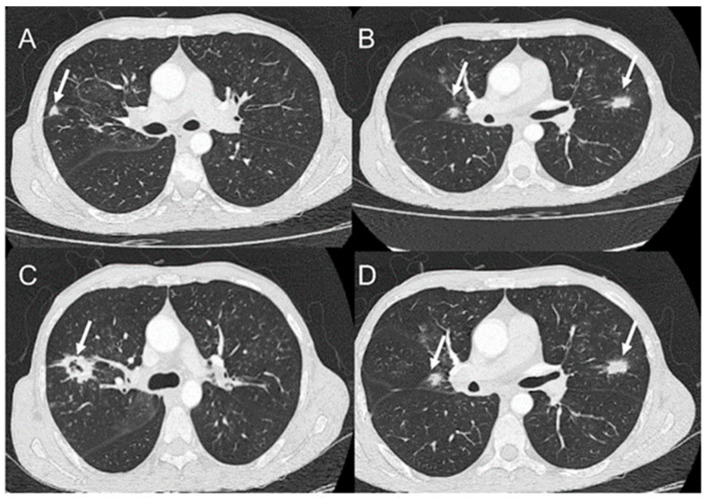
Computed tomography showing nodular opacities bilaterally distributed (arrows in **A**–**D**), with cavitation (arrow in **C**), as shown from top to bottom in figures (**A**–**D**).

**Figure 6 jof-08-01098-f006:**
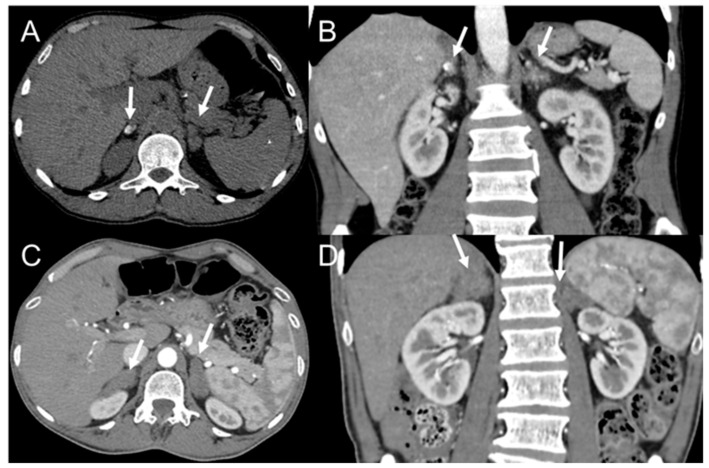
Computed tomography axial without contrast (**A**) and coronal post-contrast (**B**) images showing asymmetric adrenal thickening without significant contrast enhancement. The right adrenal is calcified, whereas the left is enlarged. Computed tomography axial post-contrast (**C**) and coronal post-contrast (**D**) from another patient showing adrenal thickening.

**Figure 7 jof-08-01098-f007:**
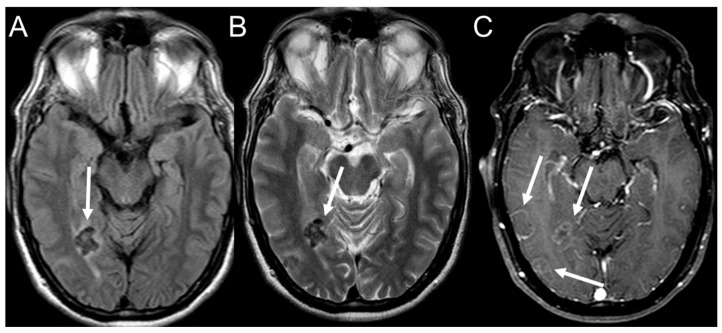
Magnetic resonance imaging showing a hypointense lesion on fluid-attenuated inversion recovery (FLAIR) (**A**) and T2-weighted imaging (T2WI) (**B**), with an annular enhancement on T1 post-contrast (**C**) located in the right occipital and temporal lobes.

## Data Availability

Not applicable.

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
