# Peer review of "Paracoccidioidomycosis: What We Know and What Is New in Epidemiology, Diagnosis, and Treatment"

_jof, 2022, doi:10.3390/jof8101098_

Round 1
Reviewer 1 Report
The manuscript is one more review on PCM. Although the title speaks about recent developments, the manuscript does not clearly show recent advances in the topic
The manuscript, in parts, appears to be written in non-technical language. It is vague based on the proposal and requires reviewing the most recent publications.
Figures should be revised, changed and better described
Line-23-25: Advances in diagnostic methods involving molecular and immunological techniques and biomarkers are promising, although they are still scarcely available outside reference centers. What progress/advances are you referring to?
Line 43-45. Discuss this point using the more global and updated reference DOI: 10.1111/myc.13510
Line 44: Which are the new antigenic methods that have been developed. Please describe and reference
Line 67-71: Include the most recent epidemiological study, carried out in South America, including the largest number of isolates. Teixeira et. al. Fungal Genet Biol. 2020;140:103395. doi: 10.1016/j.fgb.2020.103395.
Line 96: 15:1 male:female relationship indicated is maybe for in the State of Espirito Santo, Brazil (reference). Please clarify that this situation occurs in Brazil
Line 145: “most of the time” ??? write more appropriately
Line 148-149: Figure 2, including clinical manifestations is not included in this sentence
Figure 2: poor quality photos, appear old photos, do not reflect the typical clinical manifestations of PCM. These photos should not be included in a current review…
On the other hand, the clinical manifestations showed in those photos they are not described in the text. In this way they “Acute/subacute with skin lesions” or “Chronic oral lesión” … are incorrectly described
Line 149-152: A review must be more clear. Explain better and reference.
Line 165-166: Do you consider that diagnosis through secondary oral manifestations is early? please explain, discuss, or rewrite.
Line 181- 188. Clinical Diagnosis review must include not only differential diagnosis. You must talk about coinfections, usually observed in PCM. Also, COVID, what was/is the relationship observed ??? Another recent publication must be reviewed: Endemic Mycoses and COVID‑19: a Review. https://doi.org/10.1007/s12281-022-00435-z
Line 191-193: A review about PCM must discuss also the recent guidelines EORTC/ MSGERC and CMM/ISHAM.
Line 191-192. Revise this sentence. Especially “pathological methods”.
Figure 3. What do Y and FF mean, indicated on the tubes???. All images (A to D), are not well described.
Line 197-199. This phrase sounds out of place because not all types of samples are mentioned. Review the entire paragraph because the information is mixed.
Line 208: How many “agents” could be isolated?
Line 208-209: for all types of samples??
Line 210: Please include and image/photo showing ovals and elliptical Paracoccidioides mother cells
Line 216- 227. Please, review more literature or maybe discuss more clearly. In line 217-218 DME has approximately 90% positivity when … sputum samples… then you talk about the low fungal burden of the sputum.
Line 219: Paracoccidioides spp. can be mistaken with Histoplasma and Cryptococcus. In which cases? can you give examples or explain better? O show figures? It would be good for those who are going to read this review to take this situation into account
Line 229-230: …grow in the culture media, and Sabouraud and Mycosel agar. Wich culture media?? Only Sabouraud and Mycosel?? References talk about BHI for example.
Line 231: “can be” ?? must be
Line 232: If “it is advised that specific stains be used to increase detection sensitivity” why you consider first “The sections are stained with hematoxylin and eosin”.
Immunological Diagnosis/ Diagnostic Imaging. Considering the objective of the review, summarize and indicate clearly which are the recent advances.
Treatment: Talk briefly about the story or better not... Show the current situation and what are the advances.
Discuss the different Latin American situation in terms of availability of the few existing ATF. Revise Mycoses. 2022;00:1–9. DOI: 10.1111/myc.13510
Author Response
Rebuttal letter
Reviewer's comments:
Reviewer 1:
The manuscript, in parts, appears to be written in non-technical language. It is vague based on the proposal and requires reviewing the most recent publications.
After the modifications were made in response to the reviewers' questions, the article was submitted to a new evaluation by Editage. We believe that there has been an improvement in English and scientific-technical language. Moreover, in this new version, we adapted the title of the manuscript to better meet our objectives.
Figures should be revised, changed and better described.
All the figures in this manuscript were revised, and two news figures were added (Figure 6 and Figure 7).
Line-23-25: Advances in diagnostic methods involving molecular and immunological techniques and biomarkers are promising, although they are still scarcely available outside reference centers. What progress/advances are you referring to?
Done. The phrase was reformulated. Line: 23-29.
Line 43-45. Discuss this point using the more global and updated reference DOI: 10.1111/myc.13510
This point was revised, and the reference cited was inserted. Line: 44-45
Line 44: Which are the new antigenic methods that have been developed. Please describe and reference
The research has been directed to find new antigens and no new serological methods. This sentence was miswritten. Thus, it was rewritten. Line: 48-52
Line 67-71: Include the most recent epidemiological study, carried out in South America, including the largest number of isolates. Teixeira et. al. Fungal Genet Biol. 2020;140:103395. doi: 10.1016/j.fgb.2020.103395.
The description of the changes in the taxonomy of the Paracoccidioides species has been reformulated, and the most recent references have been included. Line:72-88 and 91-94.
Line 96: 15:1 male:female relationship indicated is maybe for in the State of Espirito Santo, Brazil (reference). Please clarify that this situation occurs in Brazil
We corrected the male and female ratio in chronic PCM to 15 to 22:1 and clarified that those are Brazilian references. Line: 116-117
Line 145: “most of the time” ??? write more appropriately
The discussion of pathogenesis was reformulated and inserted into the clinical diagnosis session. The expression "most of the time" has been removed. Line:167-168.
Line 148-149: Figure 2, including clinical manifestations is not included in this sentence
Figure 2 was modified, and the text’s clinical forms were more detailed. Line: 183-185, 189, 195, 221-227.
Figure 2: poor quality photos, appear old photos, do not reflect the typical clinical manifestations of PCM. These photos should not be included in a current review…
On the other hand, the clinical manifestations showed in those photos they are not described in the text. In this way they “Acute/subacute with skin lesions” or “Chronic oral lesión” … are incorrectly described
We provided new photos with better resolution and added more detailed descriptions.
Line 149-152: A review must be more clear. Explain better and reference.
The text was rewritten to make the description of pathogenesis clear. Line: 168-175
Line 165-166: Do you consider that diagnosis through secondary oral manifestations is early? please explain, discuss, or rewrite.
We corrected the information to make clear that oral lesions are not early manifestations of PCM but may lead to prompt diagnosis once they are visible and may provide samples for diagnostic studies. Line:183-185.
Line 181- 188. Clinical Diagnosis review must include not only differential diagnosis. You must talk about coinfections, usually observed in PCM. Also, COVID, what was/is the relationship observed ??? Another recent publication must be reviewed: Endemic Mycoses and COVID‑19: a Review. https://doi.org/10.1007/s12281-022-00435-z
We expanded the discussion of differential diagnosis and included a comment on PCM and covid 19. Line:214-219.
Line 191-193: A review about PCM must discuss also the recent guidelines EORTC/ MSGERC and CMM/ISHAM.
The recent guidelines were revised and inserted in the manuscript. Line: 230-233.
Line 191-192. Revise this sentence. Especially “pathological methods”.
This sentence was revised and modified as suggested. Line:233-234.
Figure 3. What do Y and FF mean, indicated on the tubes???. All images (A to D), are not well described.
The legend figure was rewritten, and on the tubes were added the fungal forms without abbreviations. Line:245-252.
Line 197-199. This phrase sounds out of place because not all types of samples are mentioned. Review the entire paragraph because the information is mixed.
The paragraph was rewritten. Line:235-237.
Line 208: How many “agents” could be isolated?
The phrase was rewritten. Line:255.
Line 208-209: for all types of samples??
The sentence was better written. Line:254-255 and 258-259.
Line 210: Please include and image/photo showing ovals and elliptical Paracoccidioides mother cells
This sentence was rewritten, and the affirmation was removed. Paracoccidioides mother cells are round. The budding cells can be oval and elliptical. Line 257.
Line 216- 227. Please, review more literature or maybe discuss more clearly. In line 217-218 DME has approximately 90% positivity when … sputum samples… then you talk about the low fungal burden of the sputum.
This paragraph was rewritten and better discussed. Line:269-275.
Line 219: Paracoccidioides spp. can be mistaken with Histoplasma and Cryptococcus. In which cases? can you give examples or explain better? O show figures? It would be good for those who are going to read this review to take this situation into account
This information was added to the manuscript. Line: 265-267.
Line 229-230: …grow in the culture media, and Sabouraud and Mycosel agar. Wich culture media?? Only Sabouraud and Mycosel?? References talk about BHI for example.
This sentence was revised and modified. Line:284-291.
Line 231: “can be” ?? must be
Accordingly, the correct form is “must be.” Line:283.
Line 232: If “it is advised that specific stains be used to increase detection sensitivity” why you consider first “The sections are stained with hematoxylin and eosin”.
We clarified the phrase, and it was poorly worded. Line:296-298.
Immunological Diagnosis/ Diagnostic Imaging. Considering the objective of the review, summarize and indicate clearly which are the recent advances.
Done. Line: 361-363 and 498-502.
Treatment: Talk briefly about the story or better not... Show the current situation and what are the advances.
We have suppressed most information related to PCM treatment history and added information about the new formulation of itraconazole (SUBA-itraconazole). Line: 514-526.
Discuss the different Latin American situation in terms of availability of the few existing ATF. Revise Mycoses. 2022;00:1–9. DOI: 10.1111/myc.13510
We clarified the current situation of the availability of antifungals in Latin America. Line:539

Reviewer 2 Report
The manuscript is well written and brings a valuable contribution in the field of medical mycoloy, specially in Paracoccidioidomycosis, the main negelcted systemic mycosis from Latin American countries.
Only two corretions needed:
Figure 2: letter D is missing in the figures
lane 185: replace "lues" for "syphilis
Figure 3: replace "(left)" for "yeast" and "(right)" for "FF"
line 354 - is it possible to write for extension FDG/PET/CT and then abbreviate?
Author Response
Reviewer 2:
The manuscript is well written and brings a valuable contribution in the field of medical mycoloy, specially in Paracoccidioidomycosis, the main negelcted systemic mycosis from Latin American countries.
Only two corretions needed:
Figure 2: letter D is missing in the figures
Figure 2 was completely modified.
lane 185: replace "lues" for "syphilis
Done. Line: 211.
Figure 3: replace "(left)" for "yeast" and "(right)" for "FF"
The suggestion was accepted.
line 354 - is it possible to write for extension FDG/PET/CT and then abbreviate?
The abbreviations were written for the extension. Line:435.

Reviewer 3 Report
Paulo Mendes-Pecanha and colleagues review recent advances in epidemiology, diagnosis and treatment of paracoccidioidomycosis. Overall the article is clear and well-structured.
1. Species and geographic distribution:
I think, it would be interesting for the readers to know, that the data found by MLST ( including mostly only two loci) were confirmed by whole genome sequencing (Mavengere et al., 2020)
Also, I would suggest including the current discussion about the etiologic agents of lobomycosis/lacaziosis. In fact, by MLST, it was shown that they cluster with the genus Paracoccidioides (Vilela et al. 2021).
2. Culture
Do you have an idea of the sensitivity of cultures? And how long does it take for paracoccdioides to grow from clinical samples?
3. Antibody detection
line 294: I think this sentence is not really clear
line 297: There was only 1 serum of P. lutzii and P. americana each. I think, you can't say: the test (based on gp43 Ab) proved to be effective for PCM diagnosis caused by P. brasiliensis sensu stricto, P. americana and P. lutzii, just based on the results of this study.
line 312: what is known about positivity of Histoplasma antigene detecion tests in patients with paracoccidioidomycosis?
line 323: what do you mean? I think you wanted to say that there is still no progress done into the availability of commercial tests for the detection of paracoccidioide antigens?
Molecular tools:
line 343: It is not only, that these tests are not validated for sputum and fresh biopsies. These are in-house tests, for which no external quality assessments are available and no consensus has been achieved for the methodology or the role these methods have in the diagnosis of PCM (nowhere). I think that they could lead to an improvement of the diagnosis, but multicenter studies with comparisons (sensitivity/specificity) are lacking, especially for specific PCRs. (Panfungal PCRs, which need subsequent sequencing are probably safer, but less sensitive, however, again, studies are lacking.
Concerning the conclusions, I would add, that there is a necessity to improve diagnostic tests (f.ex. adapt serologic tests to identify all types of PCM (especially in regard to P. lutzii), to compare the different molecular in-house tests) and to improve the accessibility to itraconazol.
Author Response
Reviewer 3.
Paulo Mendes-Pecanha and colleagues review recent advances in epidemiology, diagnosis and treatment of paracoccidioidomycosis. Overall the article is clear and well-structured.
- Species and geographic distribution:
I think, it would be interesting for the readers to know, that the data found by MLST ( including mostly only two loci) were confirmed by whole genome sequencing (Mavengere et al., 2020)
We agreed that this is a piece of important information, and it was included in the discussion. Line 84-88.
Also, I would suggest including the current discussion about the etiologic agents of lobomycosis/lacaziosis. In fact, by MLST, it was shown that they cluster with the genus Paracoccidioides (Vilela et al. 2021).
We included the new findings reported by Vilela at the end of the taxonomy discussion. Line: 91-94.
- Culture
Do you have an idea of the sensitivity of cultures? And how long does it take for paracoccdioides to grow from clinical samples?
This information was mentioned in the manuscript. Line:280-281
- Antibody detection
line 294: I think this sentence is not really clear
The sentence was clarified. Line: 353-360.
line 297: There was only 1 serum of P. lutzii and P. americana each. I think, you can't say: the test (based on gp43 Ab) proved to be effective for PCM diagnosis caused by P. brasiliensis sensu stricto, P. americana and P. lutzii, just based on the results of this study.
Accordingly, the sentence has been corrected. Line 353-360.
line 312: what is known about positivity of Histoplasma antigene detecion tests in patients with paracoccidioidomycosis?
This information was inserted into the manuscript. Line:376-377.
line 323: what do you mean? I think you wanted to say that there is still no progress done into the availability of commercial tests for the detection of paracoccidioide antigens?
Exactly. We corrected the sentence. Line: 384-388.
Molecular tools:
line 343: It is not only, that these tests are not validated for sputum and fresh biopsies. These are in-house tests, for which no external quality assessments are available and no consensus has been achieved for the methodology or the role these methods have in the diagnosis of PCM (nowhere). I think they could improve the diagnosis, but multicenter studies with comparisons (sensitivity/specificity) are lacking, especially for specific PCRs. (Panfungal PCRs, which need subsequent sequencing are probably safer, but less sensitive, however, again, studies are lacking.
Accordingly, we reformulated this text fragment. 410-426.
Concerning the conclusions, I would add, that there is a necessity to improve diagnostic tests (f.ex. adapt serologic tests to identify all types of PCM (especially in regard to P. lutzii), to compare the different molecular in-house tests) and to improve the accessibility to itraconazol.
The suggested actions were included in the conclusion and certainly are essential to improve PCM management in Latin America. Line 583-589.

Round 2
Reviewer 1 Report
Title: delete “in this disease” you are already talking about PCM
Line 72: a new classification for P. brasiliensis variants phylogenetic species
Lines 70-73-74: Phylogenetic species include S1, PS2, PS3 and PS4. Not “PS1”. Revise and correct.
Line 74. Turissini et al. doesn’t talk about S1 lineage split into two clades named S1a and S1b. Check the bibliography.
Line 76: P. brasiliensis complex
Line 172: Acute/subacute 10-15% so Chronic 85-90%. Then, Line 177: Chronic 74–96%. Line 188: Acute/subacute 5-25%. Please unify or clarify if you are talking about specific region. Remain confuse
Figure 3a: Filamentous phase and yeast phase
Figure 3c: Line 249. …different serum samples from patients with paracoccidioidomycosis are in the peripheral wells??? serum from the same patient is used in a titration. Confuse.
Figure 3D: Paracoccidioides species????. Grocott’s methenamine silver stain showing Paracoccidioides yeasts cells.
Line 255: Delete “The slides for”
Line: 281: …growth of this fungus occurs only in 85% of PCM cases. Maybe and hopefully!! that value was like that... I advise reviewing the literature because most reports speak of low sensitivity, a maximum of 65%. One of several reasons why it is promoted
declare PCM neglected. See 10.1371/journal.pntd.0007195
Line 291-292: … its low sensitivity…
Figure 4: Lines 461-462. Chest radiograph demonstrates bilateral reticular opacities in PCM, prominent in the central areas (“butterfly wing” pattern). Could you define the same patter in Computed tomography
Author Response
Note: All modifications performed in the manuscript are tracked change in green.
Reviewer 1: Title: delete “in this disease” you are already talking about PCM.
We accepted the reviewer’s suggestion.
Line 72: a new classification for P. brasiliensis variants phylogenetic species.
Done. Line 73.
Lines 70-73-74: Phylogenetic species include S1, PS2, PS3 and PS4. Not “PS1”. Revise and correct.
The manuscript was revised. Lines: 70-75
Line 74. Turissini et al. doesn’t talk about S1 lineage split into two clades named S1a and S1b. Check the bibliography.
We are completely according. We cited the correct reference. Lines: 74-75
Line 76: P. brasiliensis complex
We correct the word “complex”. Line 77
Line 172: Acute/subacute 10-15% so Chronic 85-90%. Then, Line 177: Chronic 74–96%. Line 188: Acute/subacute 5-25%. Please unify or clarify if you are talking about specific region. Remain confuse.
We decided to keep the percentage reported in the Brazilian Guidelines for PCM which is 5-25% of acute forms. We deleted the information regarding the percentage of clinical forms in the following paragraphs to avoid repetitions. Lines: 172, 175, 177, and 178.
Figure 3a: Filamentous phase and yeast phase
Done. Line: 246
Figure 3c: Line 249. …different serum samples from patients with paracoccidioidomycosis are in the peripheral wells??? serum from the same patient is used in a titration. Confuse.
We rewritten. Lines 249-250.
Figure 3D: Paracoccidioides species????. Grocott’s methenamine silver stain showing Paracoccidioides yeasts cells.
We rewritten. Lines 251.
Line 255: Delete “The slides for”
Done.
Line: 281: …growth of this fungus occurs only in 85% of PCM cases. Maybe and hopefully!! that value was like that... I advise reviewing the literature because most reports speak of low sensitivity, a maximum of 65%. One of several reasons why it is promoted.
We revised the literature, there is a variation concerning to sensitivity of culture in samples humans. Restrepo et al., for example, cite in their study a sensitivity of 80%, however, Alvarado et al. showed 86%. A study conducted by Hahn et al. found a sensitivity of 97%. One of several reasons why it is not promoted is related to the slow growth of the fungi. Line 281 and 291.
declare PCM neglected. See 10.1371/journal.pntd.0007195
Done. Line 574-577.
Line 291-292: … its low sensitivity… 290-291
We decided to change the phrase…. Despite of slowness of fungi growth…..Line: 291
Figure 4: Lines 461-462. Chest radiograph demonstrates bilateral reticular opacities in PCM, prominent in the central areas (“butterfly wing” pattern). Could you define the same patter in Computed tomography.
Yes. The same butterfly wing pattern classically described on ray-x is demonstrated in this computed tomography.
Reviewer 3 Report
line 353: . They found that of the 24 PCM-positive serum samples of patients with active 353 PCM, 100% were reactive in CIE methodology using ID Antigen®, including 11 cases of 354 infections by P. brasiliensis sensu stricto, and one by P. americana and one by P.lutzii.
line 410: However, the molecular approach is performed based on in-house tests, for which currently external quality assessments are lacking. Moreover, using molecular techniques for diagnosing disease-causing fungi directly from the clinical sample is challenging because of the complexity of DNA extraction.
Line 421 to 426: I think, that these sentences should be rather in the Discussion than in the molecular detection section.
Author Response
Note: All modifications performed in the manuscript are tracked change in green.
Reviewer 3:
line 353: They found that of the 24 PCM-positive serum samples of patients with active PCM, 100% were reactive in CIE methodology using ID Antigen®, including 11 cases of infections by P. brasiliensis sensu stricto, and one by P. americana and one by P.lutzii.
Done. Line 354
line 410: However, the molecular approach is performed based on in-house tests, for which currently external quality assessments are lacking. Moreover, using molecular techniques for diagnosing disease-causing fungi directly from the clinical sample is challenging because of the complexity of DNA extraction.
Done. Line 408-411.
Line 421 to 426: I think, that these sentences should be rather in the Discussion than in the molecular detection section.
We decided to remove this paragraph for the conclusion. Line: 582-587.